# A Comprehensive Analysis of Neoadjuvant Chemotherapy in Breast Cancer: Adverse Events, Clinical Response Rates, and Surgical and Pathological Outcomes—Bozyaka Experience

**DOI:** 10.3390/cancers17020163

**Published:** 2025-01-07

**Authors:** Cengiz Yılmaz, Baha Zengel, Orhan Üreyen, Zehra Hilal Adıbelli, Funda Taşlı, Hasan Taylan Yılmaz, Özlem Özdemir, Demet Kocatepe Çavdar, Hülya Mollamehmetoğlu, Umut Çakıroğlu, Yaşar İmren, Savaş Yakan, Enver İlhan

**Affiliations:** 1Bozyaka Education and Research Hospital, University of Health Sciences Turkey, 35170 Izmir, Turkey; bahazengel@gmail.com (B.Z.); drureyen@yahoo.com (O.Ü.);; 2Department of Medical Oncology, Izmir City Hospital, 35540 Izmir, Turkey; 3General Surgery Clinic, MedicalPoint International Hospital, Izmir University of Economics, 35575 Izmir, Turkey; 4Department of General Surgery, Izmir Faculty of Medicine, University of Health Sciences Turkey, 35540 Izmir, Turkey; 5Department of Radiology, Izmir Faculty of Medicine, University of Health Sciences Turkey, 35540 Izmir, Turkey; 6Department of Pathology, Izmir Faculty of Medicine, University of Health Sciences Turkey, 35540 Izmir, Turkey; 7Onkomer Private Oncology Radiotherapy Center, 35100 Izmir, Turkey; 8Department of Pathology, Izmir City Hospital, 35540 Izmir, Turkey; 9LaMED View Central, 35220 Izmir, Turkey; 10Nuclear Medicine Clinic, Cigli Regional Education Hospital, Bakırcay University, 35620 Izmir, Turkey

**Keywords:** breast cancer, molecular subtypes, neoadjuvant chemotherapy, adverse events, clinical progression, surgical outcomes, cavity shaving, pathological outcomes

## Abstract

This study aimed to evaluate and analyze the neoadjuvant chemotherapy (NACTx) processes and surgical and pathological outcomes in breast cancer (BC). NACTx for BC caused grade ≥ 3 adverse events in one-fifth of the patients in our study. Anthracyline cardiotoxicity (2.2%) resulted in one death (0.4%). Clinical disease progression occurred in 3.9% of the cases (14% in triple-negative BC). Cavity shaving was required in one-fifth of the patients. We found that invasive lobular carcinoma (ILC) and tumors with low SUVmax values are very risky for positive surgical margins. Axillary clinical complete response is not reliable for luminal A (LA) BC and ILC, but trustworthy for HER2-enriched and triple-negative BC. It was also found that the need for ALND decreases with HER2(+) disease and higher SUVMax values of axillary lymph nodes, but increases with ER positivity and higher PR expression levels. A pathologic complete response is unlikely in well-defined LA BC.

## 1. Introduction

Breast cancer (BC) is the most commonly diagnosed cancer and the leading cause of cancer-related deaths in women worldwide [1,2]. Although the major treatment for non-metastatic disease is surgery, neoadjuvant chemotherapy (NACTx) is frequently used to make previously unresectable tumors resectable, preserve the breast, avoid axillary dissection (AD), evaluate in vivo tumor response, and eliminate micrometastatic disease. It has become the standard treatment for locally advanced BC, but is also used in earlier stages of aggressive molecular subtypes [2,3,4,5].

The NACTx period, which lasts about 3–6 months, is challenging for patients and can lead to potentially severe and irreversible treatment-related adverse events (trAEs). Disease progression during chemotherapy (CTx) is another crucial issue that may result in a loss of eligibility for breast-conserving surgery (BCS), even mastectomy, and increase the risk of metastasis [6,7,8,9]. Upon completion of NACTx, re-imaging is carried out to assess the clinical response of the breast tumor and lymph nodes (LNs) and to decide eligibility for BCS and sentinel lymph node biopsy (SLNB). Positive surgical margins (SM) during BCS or the absence of a pathological complete response (pCR) in the axilla during SLNB may necessitate additional surgeries, prolonging anesthesia and surgical duration [10,11].

A multidisciplinary tumor board with expertise in BC care, including a breast surgeon, medical oncologist, pathologist, radiologist, and radiation oncologist, should assess patients’ eligibility for NACTx, expectations from CTx, and potential risks [12,13,14].

BC is a biologically heterogeneous disease comprising distinct molecular subtypes that significantly influence treatment responses and clinical outcomes. A review of the literature reveals a lack of standardized criteria for defining molecular subtypes, along with inconsistencies in their classification. This often results in the reporting of outcomes based on broadly defined categories, such as hormone receptor (HR)-positive or HER2-positive subtypes, which fail to account for the substantial heterogeneity within these groups. For instance, HR(+) tumors characterized by strong estrogen receptor (ER) and progesterone receptor (PR) expression, low proliferation indices, and low histologic grades typically exhibit poor responses to chemotherapy. Conversely, tumors with weaker HR expression and higher proliferative activity are generally more responsive to chemotherapy. Similarly, within HER2(+) BC, biologically distinct subgroups, such as HER2-enriched tumors and HR(+) HER2(+) tumors, can demonstrate varying degrees of chemosensitivity [15,16]. Consequently, studies based on well-defined molecular subtypes are crucial for generating more accurate, reproducible, and clinically relevant data. Such an approach is fundamental to advancing personalized treatment strategies and improving patient outcomes.

This study aimed to evaluate and analyze the treatment processes and surgical and pathological outcomes of BC patients started on NACTx after being assessed by our multidisciplinary team, seek answers to the following questions, and share the results with the literature:Rate of patients completing NACTx and undergoing surgery,Frequency of grade ≥ 3 trAEs,Clinical response rates of tumors to NACTx, which patients are risky for progression,Rates of surgeries applied to the breast and axilla,Analysis of factors affecting further surgeries in cases undergoing BCS or SLNB,Pathologic response rates based on well-defined molecular subtypes of BC.

## 2. Materials and Methods

A retrospective study was conducted on patients who received NACTx for BC between January 2015 and January 2021 at a high-volume tertiary center (Bozyaka Education and Research Hospital) in Izmir, Turkey. The demographic data, detailed baseline clinical tumor characteristics (including size, focality, histology, molecular subtype of BC, stage, result of axillary LN biopsy if performed), CTx regimens, trAEs, clinical response status of the tumors, surgeries performed, and pathologic response degrees were obtained from the patients’ medical records. A pre-NACTx BC diagnosis was made by performing a tru-cut biopsy of the breast lesion(s). The patients were clinically staged by physical examination and imaging studies. All the patients had bilateral mammography, bilateral breast, and axilla ultrasonography, and most also had breast magnetic resonance imaging (MRI) and positron emission tomography for clinical staging.

All patients beginning NACTx were examined for NACTx completion status, reasons for discontinuation if not completed, grade ≥ 3 trAEs, and whether they underwent surgery. Only patients who received at least 90% of the planned CTx and underwent surgery were included in the assessment of clinical response status and the analysis of surgical and pathological outcomes (Figure 1). Patients who did not receive adequate CTx or who did not undergo surgery or undergo delayed surgery were not included in the clinical response and surgical and pathological evaluations. The National Cancer Institute Common Terminology Criteria for Adverse Events, version 5.0, was used for grading trAEs.

### 2.1. Clinical Response Status

Based on imaging findings during (usually ultrasonography) or after CTx (both ultrasonography and breast MRI), patients were categorized into three groups: clinically responsive, unresponsive, or progressive. Tumors that decreased by at least 30% were classified as responsive, while those that increased by at least 20% were considered to have progressed. Additionally, a significant increase in the size of axillary LN(s) or the appearance of new pathologic LN(s) was also considered a progression. The findings were analyzed based on each molecular subtype of BC. The clinical courses and tumor characteristics of the patients with clinical progression were examined. Additionally, complete disappearance of axillary pathological LNs on imaging was considered a clinical complete response (cCR) in the axilla (ycN_ax_0).

### 2.2. Surgical Evaluations

The breast and axilla surgeries of patients receiving adequate CTx were evaluated. It was investigated whether patients who underwent BCS needed cavity shaving (CS) or total mastectomy (TM) and whether axillary lymph node dissection (ALND) was required in patients who underwent SLNB. The causes of ALND following SLNB (non-pCR in the lymph nodes or no lymph nodes found in the surgical specimen) were noted. BCS patients with negative SM were compared with those with positive SM needing further surgery. Patients achieving pCR at SLNB were also compared with those needing ALND due to non-pCR.

### 2.3. Pathological Evaluations

The patients were divided into five molecular subtypes based on the 2013 St. Gallen consensus criteria by examining tru-cut biopsy findings of their breast tumors: Luminal A (LA; ER strongly positive ≥ 70%, PR ≥ 20% positive, human epidermal growth factor receptor 2 [HER2] negative, and Ki-67 < 14%), luminal B-HER2 negative (other HER2 negative luminal cancers), LB-HER2(+), HER2-enriched, and triple-negative (TN) [17]. It was considered HR positive if either ER or PR was 1% or more positive. The absence of any invasive tumor (independent of DCIS) in the surgical specimen was considered pCR. Residual tumors were divided into three groups based on pathology report reviews and considering clinical tumor size: minimal residual disease (MRD < 10% residual invasive carcinoma), partially responsive tumors (tumors that show signs of regression histopathologically and decreased at least 30% in size or cellularity relative to clinical tumor size), and unresponsive tumors. Pathology preparations were examined for cases in which the response status remained unclear. SM positivity was described as ink on the tumor; in these cases, further surgeries (CS or/and TM) were performed.

### 2.4. Statistical Analysis

The distribution of variables for normality was tested using the Kolmogorov–Smirnov and Shapiro–Wilk tests. Analysis of normally and non-normally distributed quantitative data was done using an independent samples t-test and Mann–Whitney U test, respectively. Pearson’s chi-square test (*X*^2^) was used for qualitative independent data, and Fisher’s exact test was used when chi-square test conditions were not met. Univariate and multivariate logistic regression analyses were performed to investigate the effect levels of the factors on CS after BCS and ALND after the SLNB procedure. A *p*-value of less than 0.05 was considered statistically significant. The analyses were conducted using IBM^®^ SPSS^®^ statistics software version 28.0 (Armonk, NY, USA). 

## 3. Results

### 3.1. NACTx Course and Treatment-Related Adverse Events (n = 242)

The medical records of 242 patients who initiated NACTx for BC between 2015 and 2021 were reviewed. CTx-related AEs led to treatment discontinuation in eight patients (3.3%), resulting in four early operations, two delayed operations, and two unable to undergo surgery. Four patients declined to continue CTx, and one patient refused to undergo surgery (Figure 1—Section A). The remaining patients (94.6%, n = 229) received adequate CTx and underwent surgery, making them suitable for clinical response status assessment and analysis of surgical and pathological outcomes (Figure 1—Section B).

Grade ≥ 3 trAEs developed 52 times (21.5%) in 46 patients (19%) (Table 1). Docetaxel-induced hand-foot syndrome (13.7% of the patients who received docetaxel) was the most frequent (3.7%) grade ≥ 3 trAEs. Neutropenic fever requiring hospitalization developed in 6 patients (2.5%). Grade ≥ 3 anthracycline cardiotoxicity causing congestive heart failure developed in 5 out of 225 patients (2.2%), leading to one death (mortality rate = 0.4%). Two of these patients could not undergo surgery. Diabetic complications caused delayed operations in two patients.

### 3.2. Clinical Response to NACTx and Surgical and Pathological Outcomes (n = 229)

Patient and Tumor Characteristics

A total of 229 patients qualified for surgical, pathological, and clinical response status assessments. The mean age of the patients was 51 years (range 27–75), with about 85% being above 40. Most patients (89%) had BC with an invasive ductal carcinoma (IDC) histology, and 6.5% had BC with an invasive lobular carcinoma (ILC) component. When classified according to molecular subtypes of BC, 10% of the cases were LA, 45% were LB-HER2(−), 20.5% were LB-HER2(+), 8.7% were HER2-enriched, and 15.7% were TN. The predominant clinical tumor stage was T2 (63% of the patients). The majority of patients (93%) had clinically positive axillary lymph nodes (cN_ax_1–3). Axillary LN biopsy was performed in 179 patients, and 129 of them were proven to be malignant (pN_ax_+). Forty-five percent of the patients had early-stage (cT1–2 and N0–1), another 45% had locally advanced stage (cT3–4 or N2–3), and the remaining had either inflammatory or oligo-metastatic BC. Detailed tumor characteristics and NACTx regimens of the patients and a summary of the surgical and pathological outcomes are presented in Table 2.

Neoadjuvant Chemotherapy (NACTx)

The majority of the patients (88%) received sequential chemotherapy with anthracycline and taxane (AT-sCTx), while the remaining 12% received either anthracycline-based chemotherapy (Ab-CTx) or taxane-based chemotherapy (Tb-CTx) alone. The most commonly used regimens were dose-dense AC (adriamycin and cyclophosphamide, q14d), followed by weekly paclitaxel (40%), EC (epirubicin, cyclophosphamide, q21d), followed by weekly paclitaxel (20%), and dose-dense AC (q14d), followed by docetaxel (q21d). Trastuzumab alone was administered to 40% of the 67 HER2(+) patients, while 58% received dual anti-HER2 therapy (Table 2).

#### 3.2.1. Clinical Response Status

Most cases (89%) showed a notable clinical response to NACTx, while 7% (n = 16) remained unresponsive at the end of the CTx. Clinical disease progression occurred in 9 patients (Progression rate = 3.9%). Five TNBC cases progressed under Ab-CTx. Four of them continued to worsen despite switching to weekly paclitaxel, became metastatic, and died within two years. The fifth was referred for surgery, but metastatic disease developed after 26 months. Two HER2-enriched cases progressed under Ab-CTx; one was referred for surgery, and the other received taxane and dual anti-HER2 therapy, achieving a good clinical response. Two LB-HER2(−) cases also had disease progression; one of them progressed under Tb-CTx, but the other had pseudoprogression due to intratumoral necrosis and hemorrhage. The TN subtype had the highest progression rate (14%, *p* = 0.004), while the LA subtype had the highest rate of clinical unresponsiveness (30%, *p* < 0.001) (Figure 2). Eight progressed patients had IDC histology, and one had metaplastic carcinoma. The progressed patients had a higher Ki-67 index (mean: 59 ± 21 vs. 34 ± 20 and median: 65 vs. 30, *p* = 0.003) and more grade 3 tumors (78% vs. 38%, *p* = 0.031) than non-progressed patients.

#### 3.2.2. Surgical Evaluations

##### Breast Surgery

The initial surgical procedure was BCS in 59% (n = 134) of the patients. Cavity shaving was required in 30 (22%) patients, of whom 26 (19%) had an invasive tumor at the SM. Eight of them (6%) had to undergo TM. Eventually, 55% had BCS, and 45% underwent TM (Table 2).

##### Axillary Surgery

Seventy-five percent of the patients (n = 171) underwent SLNB as the initial surgical procedure due to cCR in the axilla (ycN_ax_0). However, nearly half of them (46%, n = 79) had to undergo ALND due to the presence of malignant LN(s) (n = 62) or no LN found (n = 17) in the surgical specimen. Ultimately, 40% underwent SLNB alone, and 60% underwent ALND either directly (25%) or after SLNB (35%) (Table 2).

ALND was applied at significantly higher rates in the LA (87%) and LB-HER2(−) (69%) subtypes, but at lower rates in the HER2(+) and TN subtypes (43% to 50%) (*p* = 0.001). Substantially more ALND was performed in ILC histology than IDC histology (93% vs. 57%, *p* = 0.006). Patients with cN_ax_0 had a higher SLNB rate than those with cN_ax_+ (75% vs. 37.6%, *p* = 0.032). However, patients with cN1, cN2, and cN3 disease had similar rates of SLNB (38%, 37%, and 33%, respectively). Patients with unresponsive or progressive disease and those with inflammatory or oligo-metastatic BC had very high rates of ALND (87–100%) (Table 3).

##### Risk Factors for SM Positivity During BCS

Tumor characteristics of 66 patients with clear SM and 24 patients with positive SM requiring further surgery were compared (See Figure 1, B2a; for patient selection). Patients with positive SM had substantially lower tumor SUVmax values compared to patients with clear SM (Mdn 9.4 vs. 6.2, *p* = 0.005). Compared to IDC, ILC had significantly higher positive SM rates (62.5% vs. 23.2%, *p* = 0.029). Five of the eight patients with ILC histology needed CS (Table 4). Multivariate logistic regression analysis revealed that tumor histology (IDC vs. ILC; OR: 4.962, 95% CI 1.007–24.441, *p* = 0.049) and tumor SUVMax value (OR: 0.866, 95% CI 0.755–0.993, *p* = 0.039) had significant independent efficacy on SM positivity (Table 4).

##### Factors That Increase the Likelihood of ALND After SLNB

A total of 127 patients who underwent an initial SLNB procedure due to cCR in the axilla (ycN_ax_0) were eligible for the analysis (See Figure 1, B2b; for patient selection). Of the patients, 67 had axillary pCR and did not need ALND (SLNB alone), while 60 patients underwent ALND due to non-pCR (SLNB plus ALND). There were significant differences between the molecular subtypes (*p* < 0.001). All the patients in the LA subtype who underwent SLNB required ALND. In contrast, no patient in the HER2-enriched subtype needed ALND. All the patients with ILC histology also required ALND (IDC vs. ILC; 42% vs. 100%, *p* < 0.001) (Table 5). Multivariate logistic regression analysis revealed that the ER and HER2 status, PR expression level, and SUVmax value of axillary LN(s) had significant independent efficacy on ALND requirement after SLNB. The analysis showed that ER-positive tumors had a 19 times higher probability of requiring ALND compared to ER-negative tumors. For every 10-unit increase in PR expression, there was a 17% increase in ALND probability. On the other hand, the likelihood of ALND decreased by approximately 90% in HER2-positive tumors. Additionally, each one-unit increase in the SUVmax value of the axillary LN(s) was associated with a 9% decrease in the probability of ALND (Table 5).

#### 3.2.3. Pathological Evaluations

##### General Pathologic Response (Both Breast Tumor and Axillary LNs)

Sixty-one patients (27%) achieved pCR in both the breast and the axilla. There were statistically significant differences in pathologic response status (pCR vs. non-pCR) within molecular subtypes (*p* < 0.001). No pCR was seen in the LA subtype. On the other hand, 15% of the LB-HER2(−) patients, 45% of the LB-HER2(+) patients, 65% of the HER2-enriched patients, and 33% of the TN patients achieved a pCR. Table 6 shows the general pathologic response rates by molecular subtypes and axillary lymph node status. In HER2(+) BC patients, a 30% pCR response was achieved with the combination of trastuzumab and CTx, while a 67% pCR response was achieved with a dual anti-HER2 blockade and CTx (*p* = 0.007) (Appendix A).

##### Primary Breast Tumor Pathologic Response

Sixty-five patients (28%) achieved pCR in the breast. Breast pCR rates differed significantly within molecular subtypes (*p* < 0.001). No pCR was observed in the LA subtype. On the other hand, 15% of the LB-Her2(−) patients, 51% of the LB-HER2(+) patients, 65% of the HER2-enriched patients, and 33% of the TN patients achieved pCR (Table 6 and Appendix A).

##### Axillary LN(s) Pathologic Response

Eighty-eight patients (38%) achieved pCR in the axilla (ypN_ax_0). Axillary pCR was not developed in any LA patient. However, approximately a quarter of the LB-HER2(−) patients, about half of the LB-HER2(+) and TN patients, and almost all HER2-enriched patients achieved a pCR (*p* < 0.001). Similar results were obtained in the analyses of cN_ax_+ (n = 213) and pN_ax_+ (n = 129) patients (Table 6 and Appendix A). There were 16 cN_ax_0 cases [6 LB-HER2(−), 6 LB-HER2(+), and 4 TN]. Interestingly, despite being cN0, none exhibited pCR in the axilla.

#### 3.2.4. Surgical and Pathological Outcomes Based on Molecular Subtypes

Tumor characteristics, NACTx regimens, clinical response status, and surgical and pathological outcomes of the patients according to molecular subtypes of BC are detailed in Table 7 (also see Figure 3 for surgical and pathological outcomes).

##### Luminal A (n = 23)

Almost 75% of the patients received AT-sCTx. There was no clinical progression, and 30% of the cases remained unresponsive. BCS was applied to 48%, SLNB to 13%, and ALND to 87% of the patients. CS was required in 42% of the patients. While all 17 patients who underwent SLNB required ALND (non-pCR in the axilla), only 14 underwent it (87%). No patients achieved pCR in the breast or axilla (Table 7). Also, MRD was not developed in any patient, and 39% of cases were pathologically unresponsive (Appendix A).

##### LB-HER2(−) (n = 103)

About 95% of the patients received AT-sCTx. Clinical disease progression was observed in 2% of the cases. BCS was performed on 57%, SLNB on 31%, and ALND on 69% of the patients. CS was required in 22% of the patients, and 60% (48 out of 80 patients) required ALND after SLNB. There was 16% pCR in the breast, 25% in the axilla, and 15% overall (Table 7).

##### LB-HER2(+) (n = 47)

Most patients (81%) received AT-sCTx, and 19% were given only Tb-CTx. All patients responded well to NACTx. BCS was applied to 53% of the patients, SLNB to 57%, and ALND to 43%. CS was required in 12% of the patients, and ALND following SLNB was performed in 27% of the patients. Approximately 51% pCR was obtained in the breast, 55% in the axilla, and 45% in total (Table 7).

##### HER2 Enriched (n = 20)

Most (85%) patients received AT-sCTx, 10% received Tb-CTx, and 5% (1 patient) received only Ab-CTx. Two patients (10%) progressed under Ab-CTx. One was referred for surgery, and the other was given Tb-CTx containing anti-HER2-targeting drugs. BCS was applied to 50%, SLNB to 50%, and ALND to 50% of the patients. CS was required in 20% of the patients, and ALND following SLNB was performed in 23% of the patients. There was 65% pCR in the breast, 95% in the axilla, and 65% in total (Table 7).

##### Triple-Negative (n = 36)

Approximately 92% of patients received AT-sCTx, and 8% received only Ab-CTx. Clinically progressive disease was observed in 14% of the patients (n = 5) under Ab-CTx. BCS was applied to 58% of the patients, SLNB to 56%, and ALND to 44%. CS was required in 27% of the patients, and ALND following SLNB was performed in 13% of the patients. There was 33% pCR in the breast, 47% in the axilla, and 33% in total (Table 7).

## 4. Discussion

Although NACTx has beneficial effects, especially regarding surgical outcomes, it is not an innocent treatment. Close monitoring is necessary due to the potentially severe and long-lasting side effects of chemotherapeutics and the risk of disease progression. In our study, 3.3% of patients could not complete planned NACTx due to trAEs, resulting in early surgery in 1.6%, delayed surgery in 0.8%, and ineligibility to undergo surgery in 0.8%.

### 4.1. Clinical Progression

Clinical progression rates of 3% to 4% during NACTx of BC have been reported in various studies [9,18,19]. Caudle et al. reported a 3% progression rate (59 out of 1928 patients, 1994–2007), with three patients developing distant metastases, three requiring mastectomy instead of BCS, and two becoming inoperable [9]. Nozawa et al. reported a 4% progression rate (24 out of 595 patients, 2001–2018). Disease-free survival and overall survival of the progressive patients were significantly worse than those of non-progressed patients [17]. In our research, we observed about 4% of the clinical progression, consistent with previous studies. However, progression rates varied according to BC molecular subtypes. Notably, the TN subtype exhibited the highest progression rate (14%), while no progression was observed in the LA subtype. These findings suggest that clinical progression should be assessed and monitored based on BC molecular subtypes, as shown in Figure 2.

Progressing tumors had aggressive biological features, such as higher Ki-67 index and higher tumor grade, supporting the findings of Caudle et al. [9]. Five of the thirty-six TN patients had progressed under Ab-CTx in our study. Four continued to progress despite the CTx switch. The other patient was referred for surgery. Eventually, all developed metastases and died. Two HER2(+) patients progressed under Ab-CTx. One was referred for surgery; the other received taxane and HER2-directed therapy, responding very well. In a retrospective study evaluating the prediction of radiologic progression in TNBC, 10.3% of patients (26 out of 252) who underwent NACTx demonstrated evidence of disease progression [20]. Considering that patients who progress under NACTx have worse disease-free survival and overall survival, progression in aggressive histologies like TNBC poses a significant challenge. The optimal treatment in the case of progression is unknown. If progression develops under Ab-CTx in TNBC, a CTx switch like platinum plus taxane ± immunotherapy or surgery can be considered. Based on our experience and the literature, we recommend closely monitoring TNBC patients during NACTx and referring them for immediate surgery if there is progression [18]. For HER2(+) patients who have progressed under Ab-CTx, we suggest switching to CTx regimens containing HER2-targeted drugs or starting NACTx with Tb-CTx-containing HER2-targeting drugs [21,22,23].

### 4.2. Anthracyclines

Anthracyclines are commonly used in the NACTx of BC, but they may lead to severe AEs, such as cardiotoxicity (heart failure and arrhythmias) and secondary hematological malignancies [24,25,26,27,28,29,30]. A meta-analysis of 18 studies revealed that out of 22,815 cancer patients treated with anthracyclines, 6.3% experienced clinically overt cardiotoxicity, 17.8% experienced subclinical cardiotoxicity, and 0.4% died of cardiac causes. The risk of cardiotoxicity increased with a longer follow-up over a mean period of 9 years [31]. In our study, 2.2% of patients treated with anthracyclines experienced overt cardiotoxicity during and immediately after treatment, and one patient (0.4%) died. Longer follow-ups may show higher rates of cardiotoxicity. Due to their severe side effects, it is recommended to minimize the use of anthracyclines whenever possible. Alternative treatment options include docetaxel-cyclophosphamide for the early stages of HR(+) BC and TNBC, Tb-CTx and dual HER2 targeting agents for HER2(+) disease, and carboplatin plus docetaxel for TNBC [32,33,34,35]. When anthracycline use is necessary, epirubicin can be used as an alternative to doxorubicin because of its higher cumulative dose [36].

### 4.3. LA Disease

None of the LA BC patients achieved a pCR in our study. Since most patients had cT1-T2 tumors, BCS was feasible without NACTx, but only half underwent it. Although most patients (74%) achieved a cCR in the axilla, none achieved pCR on SLNB, and all required ALND. The literature reports that pCR can occur in LA patients after NACTx. However, the definition of LA used in those studies is unclear and does not align with generally accepted criteria. Collins et al. reported 8% pCR in 114 patients, but the accepted criteria for LA disease (ER/PR positive, her2(−), Allred score > 2, no Ki-67 criterion) are not appropriate and include the LB-HER2(−) subtype [37]. Additionally, five studies in a meta-analysis included 156 LA cases and reported an axillary pCR rate of 13% [38]. However, four of the studies do not meet LA criteria [39,40,41,42]. Only one study used the 2013 St Gallen consensus criteria for LA definition and reported a pCR of 0% [17,43]. In our study, the LA criteria (ER ≥ 70, PR ≥ 20, Ki-67 < 14, and HER2 negative) are precise and suggest that achieving a pCR in the breast or axilla in patients with pure LA BC is unlikely. Administering NACTx to early-stage LA BC may lead to overtreatment and unnecessary anthracycline toxicity. Based on the ACOSOG Z0011 study results, ALND may be negligible in those who did not receive NACTx and had only 1–2 pathologic sentinel LNs removed [44]. However, when NACTx is used in these patients, unnecessary ALND is inevitable due to non-pCR. Starting with surgery or neoadjuvant endocrine therapy or non-anthracycline CTx regimens may be more appropriate in this patient group.

### 4.4. HER2(+) Disease

Studies have shown that the pCR rates improved by adding trastuzumab to NACTx for HER2(+) BC [45]. The NEOSPHERE study demonstrated that pCR rates increased from 29% to 46% when dual anti-HER2 therapy was used [46]. Our study also showed that dual anti-HER2 treatment significantly increased the pCR (67% vs. 30%, *p* = 0.007). Based on a meta-analysis, HER2-enriched patients had substantially higher pCR rates than LB-HER2(+) patients [47]. Our study also observed a trend towards higher pCR response rates in HER2-enriched patients (65% vs. 45%, *p* = 0.128). When evaluating surgical and pathological outcomes together in the LB-HER2(+) subtype, breast and axilla pCR rates (51% vs. 55%) were consistent with BCS and SLNB rates (53% vs. 57%). However, in the HER2-enriched subtype, although there were higher rates of pCR in the breast and axilla (65% vs. 95%), the rates of BCS and SLNB were lower (50% vs. 50%). The high rates of multifocal and/or multicentric disease detected in HER2(+) diseases (LB-HER2(+): 64% vs. HER2-enriched: 75%) may partially explain the necessity of TM, but it is evident that TM is performed more than necessary. All 19 HER2-enriched patients who received optimal therapy achieved a pCR in the axilla. However, unfortunately, half of them underwent ALND. The decision to proceed directly to ALND without offering SLNB in these patients may have been influenced by several factors. These may include concerns about the adequacy of SLNB after NACTx because of potential changes in lymphatic drainage, lack of standard protocols or surgeon experience in performing SLNB after systemic therapy, and reliance on pretreatment node status rather than reassessment after NACTx. Additionally, the surgeon may not be confident in the clinical complete response of axillary LNs on post-treatment radiological imaging. However, given the high pCR rates in this subtype, even if cN_ax_ positivity persists, patients should be given the opportunity for SLNB.

### 4.5. TN Disease

A 30–40% pCR can be achieved in TNBC with AT-sCTx. Adding carboplatin can increase the pCR rate to 52% [48]. Our study primarily used AT-sCTx regimens without carboplatin and achieved a pCR rate of 33%, consistent with the existing literature. The decision to administer NACTx for TNBC is often straightforward due to its pronounced chemosensitivity [48,49]. However, TNBC is not a homogeneous but a transcriptionally heterogeneous disease with distinct molecular subtypes. These subtypes include basal-like 1 (BL-1), basal-like 2 (BL-2), mesenchymal (M), and luminal androgen receptor (LAR) subtypes, each with different clinical behaviors and varying degrees of chemosensitivity. The BL-1 subtype, the most prevalent, exhibited the highest chemosensitivity, with a pCR rate of 41%. In contrast, the BL-2 and LAR subtypes are less chemosensitive, with considerably lower pCR rates (18% and 29%, respectively) [50]. Approximately one-quarter of the cases in our study showed no pathologic response, and clinical progression developed in 14% of the patients, underscoring the heightened risk of progression in this subtype. Furthermore, our data corroborate the notion that patients exhibiting progression have an unfavorable prognosis [8,19].

### 4.6. Cavity Shaving

The rate of positive SM requiring further surgeries following NACTx ranges from 5% to 40% [51]. In our study, 22% of patients needed CS and/or TM due to positive SM following BCS. Comparing patients who underwent CS due to SM positivity with those who did not can provide valuable insights regarding the surgical approach. A remarkable finding was the lower tumor SUVmax values in patients requiring CS, suggesting that tumors with lower metabolic activity could spread more diffusely, making it challenging to achieve clear SM during BCS. Additionally, a significantly higher percentage of patients with ILC required CS than those with IDC (62.5% vs. 23.2%, respectively). This highlights the unique challenges of ILC, which can spread diffusely and have less well-defined margins than IDC, making clear surgical excision more difficult. Multivariate logistic regression analysis also confirmed that both tumor histological subtype and SUVmax value play an independent role in determining the presence of an invasive tumor in the SM needing CS. ILC histology required five times more frequent CS than IDC, and tumors with higher metabolic activity needed less CS. It has been reported that additional intraoperative circumferential CS may reduce positive margins and re-excision rates in patients at high risk for positive SM [52,53].

### 4.7. ALND After SLNB

The integration and more frequent use of NACTx and SLNB in BC have significantly reduced ALND rates. In a recent study published by Tinterri et al., it was reported that cN_ax_+ patients who became ycN_ax_0 post-NACTx and underwent only SLNB had statistically significantly better recurrence-free, distant disease-free, breast cancer-specific, and overall survival rates compared to patients who became ycN_ax_0 post-NACTx but underwent ALND either directly or following SLNB [54]. The study demonstrated the importance of less invasive axillary interventions after NACTx for long-term oncological outcomes. In our study, approximately 75% of the patients achieved ycN0 in the axilla and underwent SLNB. However, half of these patients had to undergo ALND due to either no LN(s) found (22%) or non-pCR in SLNB (78%). When the independent risk factors for ALND requirement after SLNB were analyzed, ER positivity, high PR expression levels, HER2 negativity, and low SUVmax retention in axillary LN(s) were associated with an increased risk of ALND. These risk factors should be taken into consideration when planning axillary surgery in patients with cCR in the axilla following NACTx. If the possibility of ALND is thought to be high, frozen sections of the sentinel lymph nodes should be performed to avoid a second additional surgery. The analysis of patients who underwent ALND following SLNB due to lack of pCR in the axilla showed that significantly more ALND was needed in cases with ILC histology compared to cases with IDC (100% vs. 42%, *p* < 0.001). Regarding molecular subtypes, 11% of TN, 22% of LB-HER2(+), and 66% of LB-HER2(−) subtypes required ALND after SLNB. All patients in the LA subtype needed ALND, while none of the patients in the HER2-enriched subtype required it. These findings suggest that the complete clinical response status of the axilla after NACTx is not reliable for ILC histology and/or the LA molecular subtype, but is trustworthy for the HER2-enriched and TN molecular subtypes.

### 4.8. Pathological Response

In BC clinical trials, the pCR rate is the most commonly used surrogate endpoint for evaluating the effectiveness of NACTx. Previous studies have demonstrated a positive correlation between pCR and improved long-term outcomes. Patients with pCR had better disease-free survival and overall survival compared to those with residual disease. The most significant benefit was observed in patients with aggressive molecular subtypes, such as those with TNBC and HER2-positive BC [55,56,57]. Since achieving a pCR is linked to improved overall survival and reduced recurrence, various treatments are utilized or investigated in different molecular subtypes to enhance response rates. It has been shown that pCR rates increase with the addition of carboplatin and/or immunotherapy to the treatment of TNBC [58,59,60]. In our study, no patients received immunotherapy, and minimal use of carboplatin led to only one-third of patients with TNBC achieving a pCR, clearly demonstrating the need to improve treatment options. On the other hand, adjuvant therapies, such as capecitabine for triple-negative breast cancer (TNBC) and trastuzumab emtansine for HER2-positive cases, can improve long-term survival in patients with residual disease [61,62]. Pathological complete response is much less common in HR-positive HER2-negative BCs. In our study, pCR was achieved in only 15% of patients with LB-HER2(−) BC and none with LA BC. Although the pCR rates are low in these groups, NACTx or neoadjuvant endocrine therapy may help reduce tumor size, facilitate breast surgery, and minimize the extent of axillary surgery, especially in patients with smaller breasts or extensive nodal disease in the axilla.

## 5. Limitations

The main limitation of this study is its single-center, retrospective design, which may affect the generalizability of our findings. Retrospective studies face inherent challenges, including selection bias, incomplete data, and a lack of randomization. These issues can significantly impact the reliability and applicability of our results. For instance, treatment choices made in a retrospective cohort often reflect individual clinical judgment and patient preferences, resulting in varied and non-standardized treatment approaches. The single-center design of our study may limit the applicability of our findings to larger populations. Differences in institutional treatment practices, patient demographics, and resource availability across centers could significantly affect treatment choices and outcomes. For example, the generic drug containing docetaxel used in our hospital may be different from the generics used in other centers or countries and may have caused more allergic reactions or hand-foot syndrome [63]. Only trAEs observed during or just after NACTx were reported in this study. However, long-term adverse events of chemotherapeutics (such as cardiotoxicity, hematologic malignancy, or neuropathy) may develop, or the severity of an existing trAE may increase. Another limitation is that due to the nature of the study, both radiological and pathological evaluations were not standardized and performed by many physicians, causing inter-observer variability. Variability in NACTx regimens poses another challenge, as it may affect treatment responses and outcomes. The lack of standard CTx regimens and the fact that some HER2-positive patients received single-agent anti-HER2 therapy and some received dual therapy may have affected the clinical response status and surgical and pathological results. While our study aimed to capture real-world practices and outcomes, we recognize that standardizing treatment protocols might enhance the consistency of findings. Also, the relatively small number of patients, especially in HER2-enriched and LA subtypes, may hinder the generalization of the results. The lack of long-term follow-up results, such as disease-free survival and overall survival, may be viewed as another limitation. In particular, investigating the long-term outcomes of all patients with clinical progression or severe trAEs could have strengthened the study’s impact. Prospective multicenter studies, including survival analyses with more significant numbers of patients, are needed to generalize the obtained results.

## 6. Conclusions

Treatment of newly diagnosed non-metastatic BC should be planned by experienced physicians on multidisciplinary tumor boards in the field. In cases where NACTx is considered, it is essential to determine treatment targets, have a good understanding of tumor characteristics, and initiate the most appropriate treatment based on the molecular subtype of BC. It is important to note that serious AEs and disease progression may arise during treatment, so patients should be monitored carefully. Anthracycline use requires careful consideration due to its severe cardiotoxic side effects. TNBC poses the highest risk for progression. Patients should be classified based on detailed molecular subtypes rather than simply HR-positive or HER2-positive diseases. In particular, it is essential to define LA BC precisely; the potential benefit from CTx is very low in this type of cancer, even if it is locally advanced. There is a high chance of encountering positive surgical margins in tumors with ILC histology and tumors having low SUVmax values, so careful planning is essential. In LA breast cancer and ILC, cCR in the axilla is often misleading. On the other hand, in HER2-enriched BC, the probability of axillary pCR is very high, and an SLNB chance should be given even if the axilla has not completely responded to optimal NACTx. Our study was a single-center but multifaceted study that addressed the treatment process, significant side effects, clinical response, and surgical and pathological outcomes as a whole in BC patients who received NACTx. We believe that multicenter studies or meta-analyses can be conducted with a larger number of patients by expanding each step of our study.

## Figures and Tables

**Figure 1 cancers-17-00163-f001:**
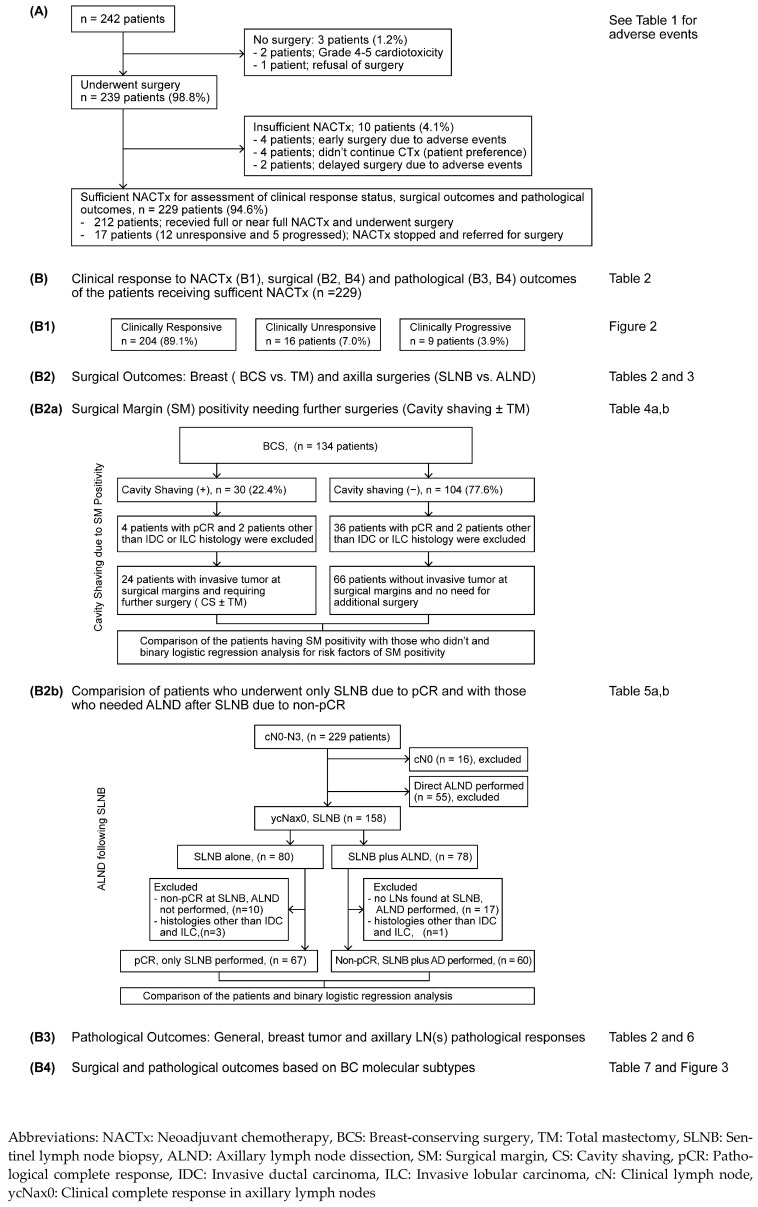
Flowchart of the study. (**A**) The clinical courses of patients started on NACTx, (**B**) clinical response status (**B1**), and surgical (**B2**,**B2a**,**b**,**B4**) and pathological outcomes (**B3**,**B4**).

**Figure 2 cancers-17-00163-f002:**
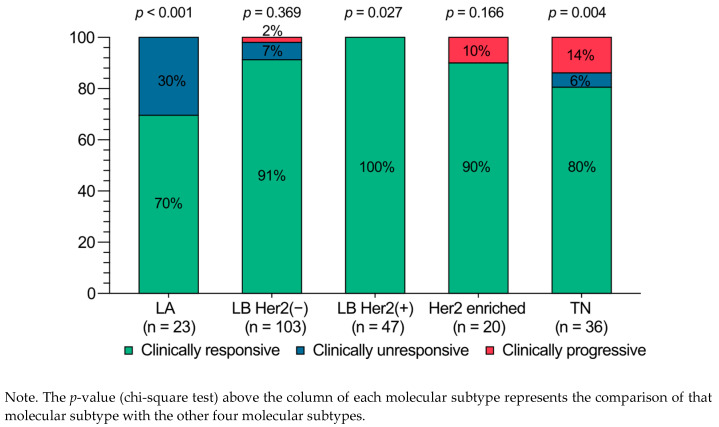
Clinical response status of tumors to NACTx based on molecular subtypes (n = 229).

**Figure 3 cancers-17-00163-f003:**
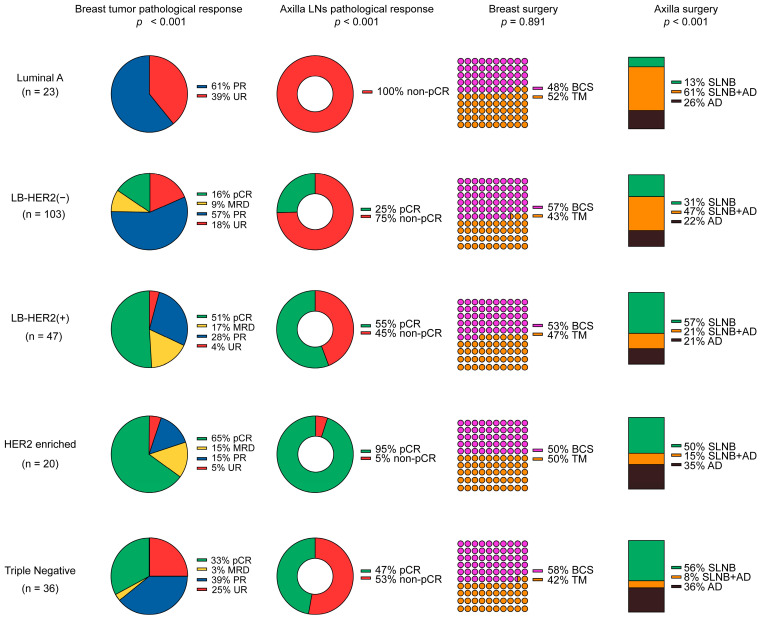
Surgical and pathological outcomes according to molecular subtypes of breast cancer.

**Table 1 cancers-17-00163-t001:** Grade ≥ 3 treatment-related adverse events (overall patients, n = 242).

Adverse Events	n	Overall	Actual *
Allergic taxane reactions (n = 225)	5	2.1%	2.2%
Anthracycline cardiotoxicity (n = 225)	5	2.1%	2.2%
Docetaxel induced hand-foot syndrome (n = 68)	9	3.7%	13.2%
Acute renal failure	3	1.2%	
Acute gastroenteritis	4	1.7%	
Paclitaxel induced hepatotoxicity (n = 154)	4	1.7%	2.6%
Neutrophenic fever	6	2.5%	
Nail toxicity	3	1.2%	
Transfusion required anemia	5	2.1%	
Peripheric neuropathy	6	2.5%	
Diabetic complications	2	0.8%	
Total adverse events in 46 patients **	52	21.5%	

Note. * The actual percentage of a drug-specific adverse event calculated according to the total number of patients receiving the corresponding chemotherapeutic agent. ** Some trAEs occurred in the same patient at different times.

**Table 2 cancers-17-00163-t002:** Clinicopathological variables, NACTx regimens, and surgical-pathological results (n = 229).

Category	Subgroups	n	%		Category	Subgroups		n	%
Age (years)	Mean ± SD (range)	51 ± 10	(27–75)		HER2 status	Positive		67	29.3
	18–30	4	1.7			(FISH positive)		(15)	(6.6)
	31–40	30	13.1			Negative		162	70.7
	41–50	84	36.7		Tumor grade	I		13	5.7
	51–60	66	28.8			II		125	54.6
	61–70	41	17.9			III		91	39.7
	>70	4	1.7		Ki-67, %	Mean ± SD (range)		35 ± 21	(5–90)
						<14		31	13.5
Menopausal status	Pre-menopausal	97	42.4			14–20		45	19.7
	Peri-menopausal	24	10.5			>20		153	66.8
	Post-menopausal	108	47.2		Axillary LN biopsy	Yes		179	78.2
Molecular subtype	Luminal A	23	10.0			No		50	21.8
	LB-HER2(−)	103	45.0		Axillary LN biopsy result (n = 179)	Malign		129	72.1
	LB HER2(+)	47	20.5			Benign		23	12.8
	HER2 enriched	20	8.7			Non-diagnostic		23	12.8
	Triple-negative	36	15.7			Suspicious		4	2.2
Clinical stage	Local (early) *	104	45.4						
	Locally Advanced **	104	45.4		**Neoadjuvant Chemotherapy**				
	Inflammatory	8	3.5						
	Oligo-metastatic	13	5.7		Cytotoxic chemotherapy regimen				
Histology	Invasive ductal	203	88.6		DD AC (q14d)-wPtx			92	40.2
	Invasive lobular	12	5.2		EC (q21d)-wPtx			47	20.5
	Mixed	3	1.3		DD AC (q14d)-Dtx (q21d)			44	19.2
	Other	11	4.8		Only taxane (T) based			15	6.6
Tumor distribution	Solitary	116	50.7		Only anthracycline (A) based			12	5.2
	Multifocal	86	37.6		Other regimens (contains A plus T)			19	8.3
	Multicentric	27	11.8						
Tumor size (mm)	Mean ± SD (range)	32 ± 14	(5–85)		Chemotherapeutic agent				
cT	T1	32	14.0		Epirubicin			56	24.5
	T2	144	62.9		Doxorubicin			158	69.0
	T3	21	9.2		Cyclophosphamide			217	94.8
	T4	32	14.0		5-Fluorouracil			4	1.7
cN_axilla_	N0	16	7.0		Docetaxel			64	27.9
	N1	125	54.6		Paclitaxel			153	66.8
	N2	73	31.9		Carboplatin			11	4.8
	N3	15	6.6						
ER status	Positive	167	72.9		Anti-HER2 drug(s) for HER2(+) disease (n = 67)				
	Negative	62	27.1		Trastuzumab			27	40.3
ER expression, %	Mean ± SD (range)	55 ± 40	(0–100)		LB-HER2(+)			(20)	
ER score	0	62	27.1		HER2 enriched			(7)	
	I	11	4.8		Trastuzumab + Pertuzumab			39	58.2
	II	21	9.2		LB-HER2(+)			(27)	
	III	135	59.0		HER2 enriched			(12)	
PR status	Positive	155	67.7		Not received			1 ***	1.5
	Negative	74	32.3						
PR expression, %	Mean + SD (range)	41 ± 39	(0–100)						
PR score	0	74	32.3						
	I	21	9.2						
	II	19	8.3						
	III	115	50.2						
**Surgery**					**Pathologic Response**				
Breast surgery	BCS	126	55.0		Breast tumor (n = 229)		pCR	65	28.4%
	Mastectomy	103	45.0				Near-pCR (MRD)	21	9.2%
Axilla surgery	SLNB	92	40.2				Partial response	103	45.0%
	ALND	137	59.8				Unresponsive	40	17.5%
Overall Surgery	PM + SLNB	64	27.9		Axillary LN(s); (cN0-N3), (n = 229)		pCR	88	38.4%
	TM + SLNB	28	12.2				Non-pCR	141	61.6%
	PM + AD	62	27.1		Axillary LN(s); (cN1-N3), (n = 213)		pCR	88	41.3%
	TM + AD	75	32.8				Non-pCR	125	58.7%
Further surgery after BCS (n = 134)	BCS (no further surgery)	104	77.6		Axillary LN(s) biopsy; malignant. pN+, (n = 129)		pCR	48	37.2%
	BCS with cavity shaving	22	16.4				Non-pCR	81	62.8%
	BCS → Mastectomy	8	6.0						
SLNB ± AD (n = 171)	SLNB only, no AD	92	40.2		Overall PR (n = 229)		pCR	61	26.6%
	SLNB, malign LN → AD	62	27.1				Partial	128	55.9%
	SLNB, no LN(s) → AD	17	7.4				Unresponsive	40	17.5%
Directly AD		58	25.3						

Note. * cT1-2 and N0-1, ** cT3-4 or N2-3, *** The progressed patient under anthracyclines-based chemotherapy. Abbreviations: SD: standard deviation, LN: lymph node, LB: luminal B, cT: clinical tumor stage, cN: clinical lymph node stage, ER: estrogen receptor, PR: progesterone receptor, EC: epirubicin + cyclophosphamide, DD: dose-dense, AC: doxorubicin + cyclophosphamide, wPtx: weekly paclitaxel, Dtx: docetaxel, q14d: every 14 days, q21d: every 21 days, BCS: breast-conserving surgery, PM: partial mastectomy, TM: total mastectomy, SLNB: sentinel lymph node biopsy, ALND or AD: axillary lymph node dissection, LN: lymph node, pCR: pathological complete response, MRD: minimal residual disease.

**Table 3 cancers-17-00163-t003:** Breast surgery (BCS vs. TM) and axilla surgery (SLNB vs. ALND) of the patients.

Category	Subgroups	Breast Surgery			Axilla Surgery		
		BCS	TM	*p*	SLNB	ALND	*p*
		55% (n = 126)	45% (n = 103)		40% (n = 92)	60% (n = 137)	
Molecular subtypes	Luminal A	48% (11)	52% (12)	0.891	13% (3)	87% (20)	0.010
	LB-Her2(−)	57% (59)	43% (44)		31% (32)	69% (71)	0.011
	LB-Her2(+)	53% (25)	47% (22)		57% (27)	43% (20)	0.011
	Her2 enriched	50% (10)	50% (10)		50% (10)	50% (10)	0.484
	Triple-negative	58% (21)	42% (15)		56% (20)	44% (16)	0.062
Histology	IDC	56% (115)	44% (89)	0.218 *	43% (87)	57% (117)	0.006 *
	ILC	40% (6)	60% (9)		7% (1)	93% (14)	
	Other	50% (5)	50% (5)		40% (4)	60% (6)	
Tumor size, mm	Median (range)	29 (5–65)	30 (8–85)	0.288	32 (7–65)	28 (5–85)	0.307 *^m^*
Focality/centricity	Solitary	74% (86)	26% (30)	<0.001	45% (52)	55% (64)	0.281
	Multifocal	43% (37)	57% (49)		41% (11)	59% (16)	
	Multicentric	11% (3)	89% (24)		34% (29)	66% (57)	
cT	cT1	59% (19)	41% (13)	0.002	44% (14)	56% (18)	0.030
	cT2	63% (90)	37% (54)		42% (60)	58% (84)	
	cT3	38% (8)	62% (13)		57% (12)	43% (9)	
	cT4	28% (9)	72% (23)		19% (6)	81% (26)	
cN	cN0	50% (8)	50% (8)	0.054	75% (12)	25% (4)	0.032
	cN1	63% (79)	37% (46)		38% (48)	62% (77)	
	cN2	44% (32)	56% (41)		37% (27)	63% (46)	
	cN3	47% (7)	53% (8)		33% (5)	67% (10)	
Clinical stage	Early	68% (71)	32% (33)	<0.001	42% (44)	58% (60)	0.006
	Locally advanced	49% (51)	51% (53)		45% (47)	55% (57)	
	Inflammatory	0% (0)	100% (8)		13% (1)	87% (7)	
	Oligo-metastatic	31% (4)	69% (9)		0% (0)	100% (13)	
NACTx	Sequential CTx	56% (114)	44% (89)	0.434	42% (86)	58% (117)	0.077
	Only A	36% (4)	64% (7)		9% (1)	91% (10)	
	Only T	53% (8)	47% (7)		33% (5)	67% (10)	
Clinical response	Responsive	57% (117)	43% (87)	0.075	44% (89)	56% (115)	0.010
	Unresponsive	44% (7)	56% (9)		13% (2)	87% (14)	
	Progressive	22% (2)	78% (7)		11% (1)	89% (8)	
Tumor pR	pCR	62% (40)	38% (25)	0.551	80% (52)	20% (13)	<0.001
	MRD	52% (11)	48% (10)		52% (11)	48% (10)	
	Partial response	54% (56)	46% (47)		28% (29)	72% (74)	
	Unresponsive	48% (19)	52% (21)		0% (0)	100% (40)	
Axillary pR	pCR	67% (59)	33% (29)	0.004	80% (70)	20% (18)	<0.001
	Non-pCR	48% (67)	52% (74)		16% (22)	84% (119)	

Note. * *p*-value calculated between IDC and ILC; other histologies were excluded. *^m^*: Mann–Whitney U Test. Abbreviations: BCS: breast-conserving surgery, TM: total mastectomy, SLNB: sentinel lymph node biopsy, ALND: axillary lymph node dissection, IDC: invasive ductal carcinoma, ILC: invasive lobular carcinoma, LB: luminal B, cT: clinical tumor stage, cN: clinical axillary lymph node stage, NACTx: neoadjuvant chemotherapy, CTx: chemotherapy, A: anthracycline, T: taxane, pR: pathological response, pCR: pathological complete response, MRD: minimal residual disease.

**Table 4 cancers-17-00163-t004:** (**a**) Comparison of BCS patients with and without further surgery (cavity shaving ± mastectomy) based on the presence of invasive tumor at surgical margins. (**b**) Binary logistic regression analysis for cavity revision due to invasive tumor at surgical margins.

**(a)**											
**Category**	**Subgroups/Unit**	**Cavity Shaving ± Mastectomy**									
		**No (SM Negative, n = 66)**				**Yes (SM Positive, n = 24)**					
		**M ± SD (Range)/n-%**			**Mdn**	**M ± SD (Range)/n-%**			**Mdn**	* **p** *	
Patient age	(years)	52	±10	(27–74)	50	50	±9	(32–66)	50	0.465	* ^t^ *
Molecular subtypes	Luminal A	7		58.3%		5		41.7%		0.544	* ^X^ * ^2^
	LB-HER2(−)	39		76.5%		12		23.5%			
	LB-HER2(+)	11		84.6%		2		15.4%			
	HER2 enriched	2		66.7%		1		33.3%			
	Triple negative	7		63.6%		4		36.4%			
Histology	Invasive ductal	63		76.8%		19		23.2%		0.029	** * ^f^ * **
	Invasive lobular	3		37.5%		5		62.5%			
Clinical Stage	Early *	37		68.5%		17		31.5%		0.307	* ^X^ * ^2-*cc*^
	Advanced **	29		80.6%		7		19.4%			
No of tumors	Solitary	40		70.2%		17		29.8%		0.520	* ^X^ * ^2-*cc*^
	Multiple	26		78.8%		7		21.2%			
cT	cT1	9		64.3%		5		35.7%		0.695	* ^X^ * ^2^
	cT2	49		75.4%		16		24.6%			
	cT3 and cT4	8		72.7%		3		27.3%			
Tumor size	(mm)	29	±11	(5–60)	29	30	±14	(8–65)	27	0.639	* ^t^ *
ER	Positive	55		74.3%		19		25.7%		0.756	** * ^f^ * **
	Negative	11		68.8%		5		31.3%			
PR	Positive	54		76.1%		17		23.9%		0.402	* ^X^ * ^2-*cc*^
	Negative	12		63.2%		7		36.8%			
HER2	Positive	13		81.3%		3		18.8%		0.544	** * ^f^ * **
	Negative	53		71.6%		21		28.4%			
ER expression	(%)	66	±35	(0–100)	80	64	±37	(0–100)	80	0.889	* ^m^ *
PR expression	(%)	48	±38	(0–100)	50	50	±40	(0–100)	60	0.945	* ^m^ *
Ki-67 index	(%)	30	±16	(7–75)	25	29	±20	(7–80)	23	0.458	* ^m^ *
Tumor grade	Low-intermediate	41		71.9%		16		28.1%		0.882	* ^X^ * ^2-*cc*^
	High	25		75.8%		8		24.2%			
Tumor SuvMax		10.1	±4.4	(2–27)	9.4	7.6	±4.7	(0–20)	6.2	0.005	* ^m^ *
(**b**)											
**Cavity Revision Required**		**Univariate Model**					**Multivariate Model**				
		**OR**	**95% CI**			** *p* **	**OR**	**95% CI**			** *p* **
Patient age		0.982	0.937	-	1.030	0.461					
Clinical stage (Early vs. advanced)		0.525	0.192	-	1.436	0.210					
Solitary vs. Multiple tumor		0.633	0.231	-	1.738	0.375					
Tumor size (mm)		1.010	0.970	-	1.051	0.635					
IDC vs. ILC		5.526	1.208	-	25.280	0.028	4.962	1.007	-	24.441	0.049
ER status (positive vs. negative)		1.316	0.405	-	4.277	0.648					
ER expression (%)		0.998	0.985	-	1.012	0.812					
ER score (0–1 vs. 2–3)		0.844	0.263		2.712	0.776					
PR status (positive vs. negative)		1.853	0.629	-	5.455	0.263					
PR expression (%)		1.002	0.990	-	1.014	0.789					
PR score (0–1 vs. 2–3)		0.625	0.233		1.679	0.351					
HER2 status (negative vs. positive)		0.582	0.150	-	2.254	0.434					
Ki-67 index		0.997	0.970	-	1.025	0.814					
Tumor grade (1–2 vs. 3)		0.820	0.307	-	2.193	0.693					
Tumor SuvMax value		0.857	0.747	-	0.982	0.026	0.866	0.755	-	0.993	0.039

Note. (a) t: *t*-test for independent samples, X^2^: chi-square test, f: Fisher’s exact test, X^2^-cc: chi-square test—continuity correction, m: Mann–Whitney U Test. (b) Logistic regression (forward LR). Abbreviations: BCS: breast-conserving surgery, M: mean, SD: standard deviation, Mdn: median, ER: estrogen receptor, PR: progesterone receptor, cT: clinical tumor stage, * cT1–2 and N0–1 and M0, ** cT3–4 or N2–3 or M1, IDC: invasive ductal carcinoma, ILC: invasive lobular carcinoma.

**Table 5 cancers-17-00163-t005:** (**a**) Comparison of clinicopathological characteristics of patients who had SLNB alone and those who underwent SLNB plus ALND. (**b**) Binary logistic regression analysis for ALND after SLNB due to non-pCR.

**(a)**												
		**Axilla Surgery**										
**Category**	**Subgroups/Unit**	**SLNB Alone (pCR, n = 67)**				**SLNB + ALND (Non-pCR, n = 60)**						
		**M ± SD (Range)/n-%**			**Mdn**	**M ± SD (range)/n-%**				**Mdn**	* **p** *	
Patient age	(years)	49	±10	(27–70)	50	51	±9	(36–75)		50	0.303	* ^t^ *
Molecular subtype	Luminal A	0		0%		13		100%			<0.001	* ^X^ * ^2^
	LB-HER2(−)	20		34%		39		66%				
	LB-HER2(+)	21		78%		6		22%				
	HER2 enriched	10		100%		0		0%				
	Triple negative	16		89%		2		11%				
Histology	IDC	67		58%		49		42%			<0.001	* ^X^ * ^2-*cc*^
	ILC	0		0%		11		100%				
Clinical stage	Early	26		42%		36		58%			0.017	* ^X^ * ^2^
	Advanced	41		63%		24		37%				
No of tumors	Solitary	36		55%		30		45%			0.674	* ^X^ * ^2^
	Multiple	31		51%		30		49%				
cT	T1	10		43%		13		57%			0.425	* ^X^ * ^2^
	T2	42		53%		38		48%				
	T3 or T4	15		63%		9		38%				
Tumor size	(mm)	34	±14	(7–65)	32	27	±13	(5–75)		25	0. 004	* ^m^ *
cN	N1	40		47%		46		53%			0.064	* ^X^ * ^2-*cc*^
	N2–N3	27		66%		14		34%				
ER status	Positive	35		38%		58		62%			<0.001	* ^X^ * ^2-*cc*^
	Negative	32		94%		2		6%				
PR status	Positive	37		40%		55		60%			<0.001	* ^X^ * ^2-*cc*^
	Negative	30		86%		5		14%				
HER2 status	Positive	31		84%		6		16%			<0.001	* ^X^ * ^2-*cc*^
	Negative	36		40%		54		60%				
ER expression	%	37	±41	0–100	10	77	±25	0–100		85	<0.001	* ^m^ *
PR expression	%	27	±33	0–100	5	67	±34	0–100		80	<0.001	* ^m^ *
Ki-67 index	%	42	±22	(8–90)	40	25	±15	(7–80)		20	<0.001	* ^m^ *
Tumor grade	Grade 1–2	32		41%		46		59%			0.002	* ^X^ * ^2-*cc*^
	Grade 3	35		71%		14		29%				
Tumor SuvMax		14	±7	(3–48)	13	9	±4	(0–23)		9	<0.001	* ^m^ *
Axilla LN SuvMax		9	±6	(0–28)	8	6	±4	(0–21)		6	0.016	* ^m^ *
(**b**)												
**ALND Required**		**Univariate Model**					**Multivariate Model**					
		**OR**	**95% CI**			* **p** *	**OR**	**95% CI**			** *p* **	
Patient age (years)		1.019	0.983	-	1.057	0.301						
Clinical stage (Early vs. advanced)		0.423	0.207	-	0.862	0.018						
Solitary vs. Multiple tumor		1.161	0.578	-	2.333	0.674						
Tumor size (mm)		0.961	0.934	-	0.989	0.007						
cN (N1 vs. N2-3-L)		0.451	0.208	-	0.976	0.043						
ER status (negative vs. positive)		26.514	5.982	-	117.515	<0.001	19.137	3.377	-	108.451	<0.001	
ER expression (%)		1.031	1.019	-	1.043	<0.001						
ER score (0–1 vs. 2–3)		10.676	3.799	-	30.002	<0.001						
PR status (negative vs. positive)		8.919	3.170	-	25.093	<0.001						
PR expression (%)		1.032	1.020	-	1.043	<0.001	1.017	1.002	-	1.032	0.029	
PR score (0–1 vs. 2–3)		8.017	3.304	-	19.454	<0.001						
HER2 status (negative vs. positive)		0.129	0.049	-	0.341	<0.001	0.110	0.035	-	0.341	<0.001	
Ki-67 index (%)		0.950	0.927	-	0.973	<0.001						
Tumor grade (grade 1–2 vs. 3)		0.278	0.129	-	0.599	0.001						
Tumor SuvMax		0.850	0.784	-	0.922	<0.001						
SuvMax value of axillary LN(s)		0.908	0.846	-	0.975	0.008	0.909	0.835	-	0.990	0.029	

Note. (a) t: *t*-test for independent samples, X^2^: chi-square test, X^2^-cc: chi-square test—continuity correction, m: Mann–Whitney U test. (b) Logistic regression (forward LR). Abbreviations: SLNB: sentinel lymph node biopsy, ALND: axillary lymph node dissection, LN: lymph node, LB: luminal B, IDC: invasive ductal carcinoma, ILC: invasive lobular carcinoma, cT: clinical tumor stage, cN: clinical lymph node stage, ER: estrogen receptor, PR: progesterone receptor, HER2: human epidermal growth factor receptor-2, pCR: pathologic complete response.

**Table 6 cancers-17-00163-t006:** General (breast and axilla), breast tumor, and axillary LN(s) pathologic responses.

	(a) Overall Patients cN_ax_(0–3); (n = 229)			(b) Clinically Axillary LN(+) cN_ax_(1–3); (n = 213)			(c) Axillary LN Bx; Malignant pN_ax_(+); (n = 129)		
	pCR	Non-pCR	*p **	pCR	Non-pCR	*p **	pCR	Non-pCR	*p **
General PR	27% (61)	73% (168)	<0.001	26% (55)	74% (158)	<0.001	26% (33)	74%	<0.001
Luminal A	0%	100%	0.005 ^cc^	0%	100%	0.006 ^cc^	0%	100%	0.012 ^f^
LB-HER2(−)	15%	85%	<0.001	13%	87%	<0.001	9%	91%	<0.001 ^cc^
LB-HER2(+)	45%	55%	0.003 ^cc^	44%	56%	0.006 ^cc^	39%	61%	0.092
HER2 enriched	65%	35%	<0.001^cc^	65%	35%	<0.001 ^cc^	71%	29%	<0.001 ^cc^
Triple-negative	33%	67%	0.433 ^cc^	34%	66%	0.327 ^cc^	46%	54%	0.094 ^f^
Breast Tumor PR	28% (65)	72% (164)	<0.001	27% (58)	73% (155)	<0.001	27%	73% (94)	<0.001
Luminal A	0%	100%	0.003 ^cc^	0%	100%	0.004 ^cc^	0%	100%	0.011 ^f^
LB-HER2(−)	16%	84%	<0.001	14%	86%	<0.001	9%	91%	<0.001 ^cc^
LB-HER2(+)	51%	49%	<0.001 ^cc^	49%	51%	<0.001 ^cc^	45%	55%	<0.018 ^cc^
HER2 enriched	65%	35%	<0.001 ^cc^	65%	35%	<0.001 ^cc^	71%	29%	<0.001 ^f^
Triple negative	33%	67%	0.606 ^cc^	34%	66%	0.442 ^cc^	46%	54%	0.113 ^f^
Axillary LN(s) PR	38% (88)	62% (141)	<0.001	41% (88)	59% (125)	<0.001	37% (48)	63% (81)	<0.001
Luminal A	0%	100%	<0.001 ^cc^	0%	100%	<0.001 ^cc^	0%	100%	0.004 ^cc^
LB-HER2(−)	25%	75%	<0.001	27%	73%	<0.001	18%	82%	<0.001 ^cc^
LB-HER2(+)	55%	45%	0.012 ^cc^	63%	37%	0.003 ^cc^	55%	45%	0.034 ^cc^
HER2 enriched	95%	5%	<0.001 ^cc^	95%	5%	<0.001 ^cc^	100%	0%	<0.001 ^cc^
Triple negative	47%	53%	0.320 ^cc^	53%	47%	0.202 ^cc^	54%	46%	0.231 ^f^

Note. *: Chi-square test, f: Fisher’s exact test, cc: continuity correction. Abbreviations: Bx: biopsy, cN_ax_: clinical axillary lymph node stage, pN_ax_: histopathologically, axillary lymph node was shown to be malignant, pCR: pathological complete response, PR: pathologic response, LN: lymph node.

**Table 7 cancers-17-00163-t007:** Tumor characteristics and surgico-pathologic outcomes according to molecular subtypes.

Categories	Subgroups/Units	Molecular Subtype										
		Luminal A (n = 23)		LB-HER2(−) (n = 103)		LB-HER2(+) (n = 47)		HER2 Enriched (n = 20)		Triple Negative (n = 36)		
		n	%	n	%	n	%	n	%	n	%	*p*
Age, years	mean ± SD	54	±10	51	±10	53	±9	49	±10	48	±11	0.028 *^a^**
Tumor size, mm	median (range)	25	(13–50)	27	(5–75)	30	(16–85)	31	(14–60)	33	(12–80)	0.097 *^kw^*
Histology	IDC	19	83%	88	85%	44	94%	19	95%	34	94%	0.166 *^X^*^2^
	ILC	4	17%	9	9%	2	4%	0	0%	0	0%	
	Other	0	0%	6	6%	1	2%	1	5%	2	6%	
Clinical stage	Early	12	52%	54	52%	18	38%	4	20%	16	44%	0.070 *^X^*^2^
	Advanced	11	48%	49	48%	29	62%	16	80%	20	56%	
cT	cT1–T2	20	87%	77	75%	38	81%	14	70%	27	75%	0.628 *^X^*^2^
	cT3–T4	3	13%	26	25%	9	19%	6	30%	9	25%	
cN	cN0–N1	18	78%	69	67%	27	58%	6	30%	21	58%	0.012 *^X^*^2^
	cN2–N3	5	22%	34	33%	20	43%	14	70%	15	42%	
Number of tumors	Solitary	13	57%	61	59%	17	36%	5	25%	20	56%	0.012 *^X^*^2^
	Multiple	10	43%	42	41%	30	64%	15	75%	16	44%	
Chemotherapy	Sequential CTx	17	74%	98	95%	38	81%	17	85%	33	92%	
	Only A	3	13%	4	4%	0	0%	1	5%	3	8%	
	Only T	3	13%	1	1%	9	19%	2	10%	0	0%	
	Trastuzumab	-	-	-	-	20	43%	7	37%	-	-	
	Dual Anti-Her2	-	-	-	-	27	57%	12	63%	-	-	
Clinical response	Responsive	16	70%	94	91%	47	100%	19	95%	29	81%	<0.001 *^X^*^2^
	Unresponsive	7	30%	7	7%	0	0%	0	0%	2	6%	
	Progressive	0	0%	2	2%	0	0%	1	5%	5	14%	
Breast surgery	BCS	11	48%	59	57%	25	53%	10	50%	21	58%	0.891 *^X^*^2^
	TM	12	52%	44	43%	22	47%	10	50%	15	42%	
Cavity shaving	No	7	58%	50	78%	23	88%	8	80%	16	73%	0.323 *^X^*^2^
	Yes	5	42%	14	22%	3	12%	2	20%	6	27%	
Breast PR	non-pCR	23	100%	87	84%	23	49%	7	35%	24	67%	<0.001 *^X^*^2^
	pCR	0	0%	16	16%	24	51%	13	65%	12	33%	
Breast surgery for cT1-T2, (n = 176)	BCS	10	50%	50	65%	23	60%	8	57%	18	67%	0.749 *^X^*^2^
	TM	10	50%	27	35%	15	40%	6	43%	9	33%	
cT3-T4, (n = 53)	BCS	1	33%	9	35%	2	22%	2	33%	3	33%	0.974 *^X^*^2^
	TM	2	67%	17	65%	7	78%	4	67%	6	67%	
Axilla surgery −1	SLNB	3	13%	32	31%	27	57%	10	50%	20	56%	<0.001 *^X^*^2^
	AD	20	87%	71	69%	20	43%	10	50%	16	44%	
Axilla surgery -2	SLNB only	3	13%	32	31%	27	57%	10	50%	20	56%	<0.001 *^X^*^2^
	SLNB plus AD	14	61%	48	47%	10	21%	3	15%	3	8%	
	Direct AD	6	26%	23	22%	10	21%	7	35%	13	36%	
Axillary LN(s) PR	non-pCR	23	100%	77	75%	21	45%	1	5%	19	53%	<0.001 *^X^*^2^
	pCR	0	0%	26	25%	26	55%	19	95%	17	47%	
Axilla surgery for cN0-N1, (n = 141)	SLNB	1	6%	25	36%	16	59%	4	67%	14	67%	<0.001 *^X^*^2^
	AD	17	94%	44	64%	11	41%	2	33%	7	33%	
cN2-N3, (n = 88)	SLNB	2	40%	7	21%	11	55%	6	43%	6	40%	0.134 *^X^*^2^
	AD	3	60%	27	79%	9	45%	8	57%	9	60%	
General pCR (breast and axilla)		0	0%	15	15%	21	45%	13	65%	12	33%	<0.001 *^X^*^2^

Note: *X*2:Chi-square test, *a*: one-way ANOVA, *** no pairwise group comparison was significant in the Bonferroni post hoc test, *kw*: Kruskal–Wallis test. Abbreviations: SD: standard deviation, IDC: invasive ductal carcinoma, ILC: invasive lobular carcinoma, cT: clinical tumor stage, cN: clinical axilla lymph node stage, LN: lymph node, BCS: breast-conserving surgery, TM: total mastectomy, PR: pathological response, pCR: pathological complete response, SLNB: sentinel lymph node biopsy, AD: axillary dissection.

## Data Availability

The datasets used and/or analyzed during the current study are available from the corresponding author upon reasonable request.

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
