# Peer review of "A Comprehensive Analysis of Neoadjuvant Chemotherapy in Breast Cancer: Adverse Events, Clinical Response Rates, and Surgical and Pathological Outcomes—Bozyaka Experience"

_cancers, 2025, doi:10.3390/cancers17020163_

Round 1
Reviewer 1 Report (Previous Reviewer 2)
Comments and Suggestions for Authors
Summary
The results of neoadjuvant chemotherapy in patients with breast cancer treated at a single facility are assessed in this publication. With a focus on molecular subtypes and predictors of axillary lymph node dissection and cavity shaving, it offers a thorough examination of clinical reactions, adverse events, and surgical and pathological outcomes.
Strengths
The study provides a detailed dataset, covering a wide range of clinically relevant metrics and outcomes for breast cancer patients undergoing NACTx.
The analysis stratified by molecular subtypes highlights the heterogeneous responses to NACTx, which is important for personalized treatment.
The insights into ALND predictors and cavity shaving implications are valuable for refining surgical strategies.
Recommendations
Figure 1's incredibly low resolution makes it challenging to analyze the data. A high-resolution version is necessary for accurate assessment.
To increase clarity and visual appeal, the figure should be redone with distinct labeling and unambiguous annotations.
It is difficult to derive significant lessons from Tables 1–7 due to their overwhelming size and poor organization. Would the authors think about combining the tables or using graphical representations like heatmaps, bar charts, or summary plots to highlight the main conclusions?
For example, rather than being dispersed across several tables, subgroup comparisons between subtypes could be shown in a single comparative table or figure.
Although pCR rates are included in the manuscript, there is insufficient discussion of how they affect survival and recurrence risk. Could the writers provide more details about how pCR rates affect long-term results and choices?
Not enough attention is paid to the drawbacks, which include the retrospective design and the variation in treatment regimens. Could you specifically discuss the potential impact of these factors on the outcomes?
Patients who test positive for HER2 are not sufficiently discussed. In certain instances, ALND was still necessary despite high pCR rates. Could the writers provide a more thorough explanation of this contradiction?
Although they are mentioned, TNBC progression rates are not thoroughly examined. What tactics might slow down these high-risk patients' advancement during NACTx?
Conclusion
With an emphasis on molecular subtype variations, this paper offers a thorough examination of NACTx in breast cancer. However, its clarity and power are diminished by low-quality figures, overly complex and badly organized tables, and a lack of investigation into its limitations. Resolving these problems will greatly improve the paper's quality and usefulness, as will improved visualization and a more thorough examination of the consequences of pCR. It will make a significant contribution to the field after it has been revised.
Author Response
Dear Reviewer,
We would like to thank you for your insightful comments and suggestions. We made all possible changes that were suggested and detailed the changes in the table below. Prior to responding to your comments, we want to inform you that all the revisions and improvements are highlighted in red in the revised version of our manuscript. We sincerely appreciate your insightful comments on our paper. We would like to thank you again for your valuable time and insight in strengthening our paper.
Yours truly,
The corresponding author on behalf of the authors.
Summary of all Revisions : The title was shortened. The introduction was improved, and new citations were added (references 15-16). Materials and Results, especially regarding tables and figures, were improved. Figure 1 was made clear and visible. P values in Figure 2 were explained. A new figure (figure 3) was added to summarize surgical and pathological outcomes. Tables 4a and 4b were combined, as well as 5a and 5b. Table 6 a/b/c were collected in one table as Table 6. The discussion was improved. TNBC and HER2 disease were discussed in detail, and a pathological response section was added to the discussion section. Limitations were improved. Abbreviations were added before the references.
Comment 1. Figure 1's incredibly low resolution makes it challenging to analyze the data. A high-resolution version is necessary for accurate assessment.To increase clarity and visual appeal, the figure should be redone with distinct labeling and unambiguous annotations.
Response 1. [Figure 1 and other figures were improved]. Thank you for pointing this out. We agree with this comment. Therefore, we received professional assistance from MDPI author services to improve the quality of the figures and article layout. The figures were re-made at higher resolution. Figure 1 on page 4, Figure 2 on page 9, Figure 3 on page 15.
Comment 2. It is difficult to derive significant lessons from Tables 1–7 due to their overwhelming size and poor organization. Would the authors think about combining the tables or using graphical representations like heatmaps, bar charts, or summary plots to highlight the main conclusions? For example, rather than being dispersed across several tables, subgroup comparisons between subtypes could be shown in a single comparative table or figure.
Response 2. [Tables were re-designed, Some were combined, and a new figure was added ]. Thank you for pointing this out. We agree with this comment. Therefore following revisions were done.
- All the tables were re-designed.
- Tables 4a -4b were combined. ( Page 11, lines 271-274)
- Tables 5a -5b were combined. (Pages 12-13, lines 291-294)
- Tables 6a, 6b, and 6c were combined under Table 6 (Page 14, lines 318 – 319).
- Detailed tables (6a-6b-6c) were provided as a Supplementary file (Table S1).
- Table 7 (Surgical and Pathological outcomes according to molecular subtypes of BC ) was visualized as Figure 3 ( extracted from a supplement). Page 15, lines 355-357.
Comment 3. Although pCR rates are included in the manuscript, there is insufficient discussion of how they affect survival and recurrence risk. Could the writers provide more details about how pCR rates affect long-term results and choices?
Response 3. [A new section, “4.8. Pathological Response,” was written in the discussion]. Thank you for pointing this out. We agree with this comment. Therefore, a new section was written in the discussion under the name of “ 4.8 pathological response” on page 20, lines 512-532. The importance of pathological response, how to increase pathological response, and approach to patients with pathological incomplete response were discussed together with our study results and literature. New citations were added ( references 55-62)
Comment 4. Not enough attention is paid to the drawbacks, which include the retrospective design and the variation in treatment regimens. Could you specifically discuss the potential impact of these factors on the outcomes?
Response 4. [Limitations section was improved ]. Thank you for pointing this out. We agree with this comment. Improvements were made to the limitations section regarding the single-center nature of the study, its retrospective design, and the variety of treatment regimens on pages 20-21, lines 534-544, 550-551, 554-556.
Comment 5. Patients who test positive for HER2 are not sufficiently discussed. In certain instances, ALND was still necessary despite high pCR rates. Could the writers provide a more thorough explanation of this contradiction?
Response 5. [Discussion about HER2+ breast cancer was improved ]. Thank you for pointing this out. We agree with this comment. Therefore, in the “HER2(+) disease section” of the discussion (4.4.), it was discussed why directly ALND could have been performed and SLNB was not performed in these patients, despite the development of a pathological complete response in the axilla, on pages 18, lines 444-451.
Comment 6. Although they are mentioned, TNBC progression rates are not thoroughly examined. What tactics might slow down these high-risk patients' advancement during NACTx?
Response 6. [Discussion about triple-negative breast cancer progression was improved ]. Thank you for pointing this out. We agree with this comment. Therefore, an article (reference 20) on the frequency of clinical progression in triple-negative breast cancer was added to the “clinical progression” section (4.1.) of the discussion. The approach to triple-negative patients who progress under chemotherapy was discussed on page 17, lines 382- 389.

Reviewer 2 Report (Previous Reviewer 4)
Comments and Suggestions for Authors
This manuscript is indeed a well-arranged and elaborate study of the outcomes in BC patients undergoing neoadjuvant chemotherapy treatment. It confers good insight into heterogeneity at the level of molecular subtypes and, further, at the unequal response to NACTx, underlining the most relevant clinical and pathological results in terms of progression rates, types of surgery performed, and also pCR rates. Integration of AE analysis, subtype-specific outcomes, and surgical challenges in this study makes it comprehensive; therefore, it is useful both for the clinicians and the researchers.
Author Response
Dear Reviewer,
We sincerely thank you for taking the time to review our article and for your positive evaluation. We are truly honored that you found our work worthy towards acceptance without any suggestions for revision. Your support and recognition mean a great deal to us, and we deeply appreciate your encouraging feedback. Thank you once again for your time and for accepting our article.Yours truly,
The corresponding author on behalf of the authors.
Comments and Suggestions for Authors: This manuscript is indeed a well-arranged and elaborate study of the outcomes in BC patients undergoing neoadjuvant chemotherapy treatment. It confers good insight into heterogeneity at the level of molecular subtypes and, further, at the unequal response to NACTx, underlining the most relevant clinical and pathological results in terms of progression rates, types of surgery performed, and also pCR rates. Integration of AE analysis, subtype-specific outcomes, and surgical challenges in this study makes it comprehensive; therefore, it is useful both for the clinicians and the researchers.
No suggestion was reported. It has been marked that the introduction section could be improved. In the introduction section, it was emphasized that breast cancer exhibits considerable heterogeneity, making the accurate definition of its molecular subtypes crucial especially for treatment purposes on pages 2-3, lines 83-98.

Reviewer 3 Report (Previous Reviewer 1)
Comments and Suggestions for Authors
Unfortunately, the authors did not provide responses to the reviewers' comments at the previous stage of review. 1) I still have a comment about the long title, I think it is necessary to shorten it. 2) Figure 2 shows p-values, please explain for which pair of subtypes the values ​​are. Overall, the manuscript has improved, the authors have made quite serious changes.
Author Response
Dear Reviewer,
We would like to thank you for your insightful comments and suggestions. We made all possible changes that were suggested and detailed the changes in the table below. Prior to responding to your comments, we want to inform you that all the revisions and improvements are highlighted in red in the revised version of our manuscript. We sincerely appreciate your insightful comments on our paper. We would like to thank you again for your valuable time and insight in strengthening our paper.
Yours truly,
The corresponding author on behalf of the authors.
Summary of all Revisions : The title was shortened. The introduction was improved, and new citations were added (references 15-16). Materials and Results, especially regarding tables and figures, were improved. Figure 1 was made clear and visible. P values in Figure 2 were explained. A new figure (figure 3) was added to summarize surgical and pathological outcomes. Tables 4a and 4b were combined, as well as 5a and 5b. Table 6 a/b/c were collected in one table as Table 6. The discussion was improved. TNBC and HER2 disease were discussed in detail, and a pathological response section was added to the discussion section. Limitations were improved. Abbreviations were added before the references.
Comment 1. I still have a comment about the long title, I think it is necessary to shorten it.
Response 1. [Title was shortened]. Thank you for pointing this out. We agree with this comment. Therefore, The title of the article was shortened for the second time on page 1, lines 2-4. However, the title has been shortened to a limited extent because it highlights that the content of the Study is rich.
Comment 2. Figure 2 shows p-values, please explain for which pair of subtypes the values ​​are.
Response 2. [p values were explained]. Thank you for pointing this out. We agree with this comment. Therefore, It is indicated as a note at the bottom of Figure 2, on page 9, lines 244-245. “Note. The p-value (chi-square test) above the column of each molecular subtype represents the comparison of that molecular subtype with the other four molecular subtypes.”

This manuscript is a resubmission of an earlier submission. The following is a list of the peer review reports and author responses from that submission.
Round 1
Reviewer 1 Report
Comments and Suggestions for Authors
I liked the manuscript. The article describes in detail the entire process of neoadjuvant chemotherapy (NACT) in breast cancer (BC), its significant treatment-related adverse events (trAEs), clinical response rates of tumors, and surgical and pathological outcomes, as well as analyze factors influencing further surgeries like cavity shaving and sentinel lymph node biopsy (SLNB) plus axillary lymph node dissection (ALND). The study design does not raise any questions, the material is presented logically and consistently, the discussion is detailed and thorough. In my opinion, there are too many tables with a complex structure. But the described material cannot be presented in any other form, including in the form of figures.
I would recommend that the authors add a reminder scheme in the Discussion section, which possible factors should be taken into account for each molecular biological subtype of breast cancer during neoadjuvant chemotherapy.
Minor comments:
1. The manuscript title is very long. Abbreviations in titles are not welcome, if you decipher them, the title will become even heavier. I would recommend the authors to shorten, for example, "A Comprehensive Analysis of Neoadjuvant Chemotherapy in Breast Cancer: Bozyaka Experience".
2. Table 1 indicates Total (46 patients), in the next column n=52. How should this be understood? Does it mean that several treatment-related adverse events were observed in the same patients?
3. Figure 2 - the p-value is indicated for comparing which subgroups?
Reviewer 2 Report
Comments and Suggestions for Authors
Summary
Neoadjuvant chemotherapy in breast cancer is evaluated in this publication with an emphasis on adverse events, clinical responses, surgical results, and factors that affect the need for additional procedures. It highlights variations in results by subtype, such as cavity shaving in breast-conserving surgery, axillary lymph node dissection needs, and pathologic complete response rates. The study uses information from a single Turkish center's extensive retrospective analysis.
Strengths
The study uses a sizable cohort of 229 patients, providing robust data for evaluating NACT outcomes in breast cancer.
The analysis distinguishes between molecular subtypes, offering valuable insights into their differential responses to treatment.
It thoroughly discusses surgical outcomes, including BCS, mastectomy, ALND, and cavity shaving rates.
The study employs appropriate statistical methods, including logistic regression, to identify predictors of surgical outcomes.
Recommendations
Why not provide examples or interpret the implications of high PR expression and ER positivity on surgical decisions?
How about using Kaplan-Meier plots or heatmaps for a more intuitive representation of survival or subtype-specific responses?
Where is the actionable information for clinicians deciding on NACT protocols? Why not suggest specific implications for HER2-enriched versus luminal subtypes?
Why is there no inclusion of ROC curves or other diagnostic metrics to evaluate the predictive accuracy of identified factors?
The authors acknowledge the study’s single-center design but fail to discuss how this impacts generalizability. Why not address potential biases due to local protocols or population differences?
Conclusion
The paper offers an abundance of information on NACT in breast cancer, but its excessively technical presentation and lack of useful insights limit its therapeutic use. The study's importance could be greatly increased by updating the figures, concentrating on prognostic indicators, and offering more precise clinical implications. Resolving these problems will increase the findings' influence and accessibility for researchers and physicians alike.
Reviewer 3 Report
Comments and Suggestions for Authors
The article is based on a conventional scenario, and the key elements of this work have been identified. The aforementioned facts and their analysis follow a logical sequence and are undeniable.
Reviewer 4 Report
Comments and Suggestions for Authors
This manuscript is a comprehensive retrospective analysis of breast cancer patients treated with neoadjuvant chemotherapy at a high-volume tertiary center for clinical progression, treatment-related adverse events, and surgical and pathological outcomes. Notably, the comprehensive look into molecular subtypes conducted by the authors is commendable. These results give valuable clinical insight into the differential rates of pCR and nuanced recommendations for HER2-enriched, triple-negative, and luminal A breast cancers. The authors employed multivariate logistic regression to identify independent predictors of surgical margin status and necessity of ALND, further enhancing the scientific rigor of this analysis.
Limitations that should be addressed before publication include the following:
Latest Evidence of SLNB after neoadjuvant chemotherapy: The authors should discuss more about the latest evidence after sentinel lymph node biopsy in cN+ breast cancer patients after neoadjuvant chemotherapy. In fact, patients treated with less invasive axillary surgery apparently present better long-term oncological outcomes. Please cite PMID: 39335140 to improve the quality of your manuscript.
Study Design and Bias: This investigation is a single-center retrospective study, hence should be pervious to selection and treatment bias. This limitation should be fully acknowledged by the authors, and its implications should be discussed.
Response Definition: Although well-defined criteria of response for tumors are given, elaboration of interobserver variability and details of imaging applied would serve to further enhance reproducibility.
Incomplete Follow-Up Data: Long-term outcomes-for example, disease-free survival and overall survival of the patients, in particular those showing clinical progression or severe toxic effects, are missing.
Small Sample Subgroups: The results for some subgroups, including HER2-enriched (n=20) and luminal A (n=23), have very limited samples that undermine the generalizability of some specific results.